# Exome sequencing identifies novel genetic variants associated with varicose veins

**Dan-Dan Zhang[1]☯, Xiao-Yu He[2]☯, Liu Yang[2]☯, Bang-Sheng Wu[2]☯, Yan Fu[1], Wei-Shi Liu[2], Yu Guo[2], Chen-Jie Fei[2], Ju-Jiao Kang[3,4], Jian-Feng Feng[3,4,5], Wei Cheng[2,3,4,5], Lan Tan[1]\*, Jin-Tai Yu[2]\***

1 Department of Neurology, Qingdao Municipal Hospital, Qingdao University, Qingdao, China, 2 Department of Neurology and National Center for Neurological Disorders, Huashan Hospital, State Key Laboratory of Medical Neurobiology and MOE Frontiers Center for Brain Science, Shanghai Medical College, Fudan University, Shanghai, China, 3 Institute of Science and Technology for Brain-inspired Intelligence, Fudan University, Shanghai, China, 4 Key Laboratory of Computational Neuroscience and Brain-Inspired Intelligence, Fudan University, Ministry of Education, Shanghai, China, 5 Department of Computer Science, University of Warwick, Coventry, United Kingdom

☯ These authors contributed equally to this work.
\* dr.tanlan@163.com (LT); jintai_yu@fudan.edu.cn (J-TY)

**Data Availability Statement:** The main data, including individual-level phenotype and sequencing data, used in this study were accessed from the UK Biobank under application number 19542 (https://www.ukbiobank.ac.uk/). Statistics

## Abstract

### Background

Varicose veins (VV) are one of the common human diseases, but the role of genetics in its development is not fully understood.

### Methods

We conducted an exome-wide association study of VV using whole-exome sequencing data from the UK Biobank, and focused on common and rare variants using single-variant association analysis and gene-level collapsing analysis.

### Findings

A total of 13,823,269 autosomal genetic variants were obtained after quality control. We identified 36 VV-related independent common variants mapping to 34 genes by single-variant analysis and three rare variant genes (*PIEZO1*, *ECE1*, *FBLN7*) by collapsing analysis, and most associations between genes and VV were replicated in FinnGen. *PIEZO1* was the closest gene associated with VV ($P = 5.05 \times 10^{-31}$), and it was found to reach exome-wide significance in both single-variant and collapsing analyses. Two novel rare variant genes (*ECE1* and *METTL21A*) associated with VV were identified, of which *METTL21A* was associated only with females. The pleiotropic effects of VV-related genes suggested that body size, inflammation, and pulmonary function are strongly associated with the development of VV.

### Conclusions

Our findings highlight the importance of causal genes for VV and provide new directions for treatment.

for varicose veins in FinnGen are available on the website (http://r8.finngen.fi) and are indicated with the phenotype code I9_VARICVE. The scRNA-seq data used in the present study were acquired from GEO database (https://www.ncbi.nlm.nih.gov/geo/) with the accession number: GSE201333. The following software and packages were used for data analysis: FUMA v.1.3.8 (https://fuma.ctglab.nl/), SnpEff (https://pcingola.github.io/SnpEff/), GCTA v.1.93 (https://yanglab.westlake.edu.cn/software/gcta/#COJO), SAIGE-GENE+ (https://github.com/saigegit/SAIGE), BHR (https://github.com/ajaynadig/bhr), PLINK (https://www.cog-genomics.org/plink/) and R v.4.2.0 (https://www.r-project.org/), DAVID, (https://david.ncifcrf.gov/), Gene-SCOUT (https://astrazeneca-cgr-publications.github.io/gene-scout/). Scripts used to perform the analyses are available at https://github.com/ddzhang877/vv_wes.

**Funding:** JT, Yu was supported by grants from the Science and Technology Innovation 2030 Major Projects (2022ZD0211600), National Natural Science Foundation of China (82071201, 82071997), Shanghai Municipal Science and Technology Major Project (2018SHZDZX01), Research Start-up Fund of Huashan Hospital (2022QD002), Excellence 2025 Talent Cultivation Program at Fudan University (3030277001), and Shanghai Talent Development Funding for The Project (2019074). W Cheng was supported by grants from the Shanghai Rising-Star Program (21QA1408700). JF Feng was supported by grants from the 111 Project (B18015). The funders had no role in study design, data collection and analysis, decision to publish, or preparation of the manuscript.

**Competing interests:** The authors have declared that no competing interests exist.

## Author summary

In this study, whole-exome sequencing data from the UK Biobank were used to explore the effect of genetic variants on varicose veins (VV) and search for new VV-related genes. In contrast to traditional association studies, large-scale whole-exome sequencing analysis is more capable of identifying rare genetic variants (MAF < 1%) in diseases. The current study identified 34 VV-associated common variant genes by single-variant analysis and three rare variant genes (*PIEZO1*, *ECE1*, *FBLN7*) by collapsing analysis, and most associations were validated in FinnGen. In addition to replicating several genes reported in previous genome-wide association studies, we identified 17 novel genes that may be associated with VV. Through subsequent phenome-wide association analyses of identified genes, we found that these genes are also strongly associated with body size, inflammation, and pulmonary function. These findings contribute to understanding the underlying mechanisms of pathogenesis and developing novel therapeutic strategies for VV.

## Introduction

Varicose veins (VV) are the most common chronic condition of the venous system, often affecting the lower extremities and manifesting as dilated, stretched, or tortuous superficial veins [1]. The majority of VV patients suffer from complications such as pain, swelling, hyperpigmentation, and ulcers. About 23% of adults aged 40–80 in the United States develop VV, including 22 million women and 11 million men [2]. Today, endovenous laser ablation is considered clinically the first-line treatment option for VV but has a post-operative recurrence rate of up to 20% [3,4]. Therefore, it is particularly important to define the etiology of VV. Previous epidemiological studies have shown that VV are associated with several risk factors, including advanced age, being female, pregnancy, obesity, height, and history of deep vein thrombosis [5–8]. A positive family history is one of the important risk factors for VV, which suggests that VV are likely to be modulated by genetic factors [9–12]. Most of the previous genetic studies on VV were genome-wide association studies (GWAS) [13–15]. Within the last decade, dozens of genomic loci associated with VV have been identified. However, GWAS are more limited in scope and mostly focused on common variants (minor allele frequency (MAF > 1%)), which usually have a small effect size and cannot directly identify the causal gene.

In contrast to traditional studies, large-scale whole-exome sequencing (WES) analysis is more capable of identifying rare genetic variants (MAF < 1%) in diseases [16]. Rare variants are genetic markers of high disease risk and help to identify novel genetic targets for drug interventions [17]. Apart from revealing the full spectrum of protein-coding variants, WES can facilitate the identification of novel loss-of-function (LOF) variants and enhance the ability to detect the associations between LOF variants and diseases. LOF variants can identify drivers of genetic risk, novel disease genes, and therapeutic targets [16]. Also, large-scale sequencing can evaluate the disease prevalence and carrier frequencies of rare genetic variants [18]. A recent study focused on the effect of *PIEZO1* on altered VV risk and identified the association of VV with rare protein-truncating variants in *PIEZO1* [19]. A multi-phenotype exome-wide association study (ExWAS) from the UK Biobank (UKB) involving 49,960 participants [16] identified novel LOF variants with significant disease impact, including *PIEZO1* on VV ($P = 3.2 \times 10^{-8}$). In addition, a phenome-wide level ExWAS [20] exploring the contribution of rare variants to human disease also found the association between *PIEZO1* and VV ($P = 3.24 \times 10^{-24}$). In contrast to their studies, we conducted a more comprehensive ExWAS of

VV using the updated exome sequencing data of more than 350,000 UKB individuals, with additional validation and exploration of the findings and biological functions. We identified 19 known and 17 novel causal genes significantly associated with VV, with most of these associations validated in FinnGen [21]. In addition, our study explored the phenotypic associations between VV genes and multiple traits, showing that cardiovascular disease, height, biochemistry, and inflammatory indicators might contribute to the development of VV.

## Results

### Participant characteristics

In this study, we performed an ExWAS by utilizing phenotypic and genetic data from UKB, including exome sequencing data and VV diagnosis data (**S1 Table**). Following stringent quality control (QC) of genotypes and samples, we identified 350,770 unrelated Caucasian participants (21,643 VV cases and 329,127 controls) with a mean age of 56.94 years at enrollment, of which 53.76% were female (**S2 Table**). The case and control groups had similar age distribution (**S1 Fig**). The clustering of the VV case and control groups in the principal component analysis is shown in **S2 Fig**. A total of 13,823,269 autosomal genetic variants were obtained after QC, including 100,098 common variants (MAF > 1%) and 13,723,171 rare variants (MAF < 1%).

### Exome-wide single-variant association analysis for VV

The mixed-effects logistic models adjusted for age, sex, and the top 10 principal components (PCs) were used to assess the association between VV and common coding variants. Firstly, 169 variants and 114 variants were found to reach the exome-wide significance ($P < 1.19 \times 10^{-6}$) and the genome-wide significance ($P < 5 \times 10^{-8}$) in the single-variant analysis (**Fig 1A** and **S3 Table**). After clumping 169 variants in strong linkage disequilibrium, 36 independent common variants associated with VV were identified (**Fig 1A**), of which 34 were successfully validated in FinnGen [21] (**Table 1** and **S4 Table**). Among these independent common variants, 19 were significantly protective against VV, while the other 17 were significantly associated with an elevated risk of VV (**Table 1**). In addition, 36 independent common variants were mapped to 34 genes, of which 18 have been reported previously and the other 16 genes were novel (including *TRIM10, UBE2H, TUBAL3, DUSP8, DNAH10, CAPRIN2, MSL1, ZBTB4, CIB3, SERPIND1, SHANK3, REST, CTXN3, H2AC6, PGBD1, ABHD16A*). *PNO1* and *PIEZO1* had the most significant associations with VV, and the *P*-values were $9.81 \times 10^{-44}$ and $1.75 \times 10^{-39}$, respectively. *TRIM10* showed the strongest association with VV among 16 novel common variant genes ($P = 5.29 \times 10^{-14}$). The quantile-quantile (Q-Q) plot of single common variants is shown in **Fig 1B**. It was noteworthy that these independent common variants were characterized by larger effect sizes at lower frequencies (**Fig 1C**). Furthermore, we conducted a conditional analysis of the identified common variants linked to VV using conditional and joint analysis (COJO). The results revealed that 27 independent common variants were retained, 25 of which were consistent with variants identified after clumping (**S5 Table**).

### Three VV genes identified by rare variant collapsing analysis

We performed the gene-level collapsing analysis to determine rare variant associations with VV. To elucidate the differences in genetic structures between genes, our analysis included 19,897 genes under 8 different models (**Methods**). Collapsing analysis showed 13 significant associations involving 3 genes (*PIEZO1, ECE1, FBLN7*) after Bonferroni correction ($P < 2.51 \times 10^{-6}$) (**Fig 2A** and **S6 Table**). The corresponding Q-Q plots are shown in **S3 Fig**.

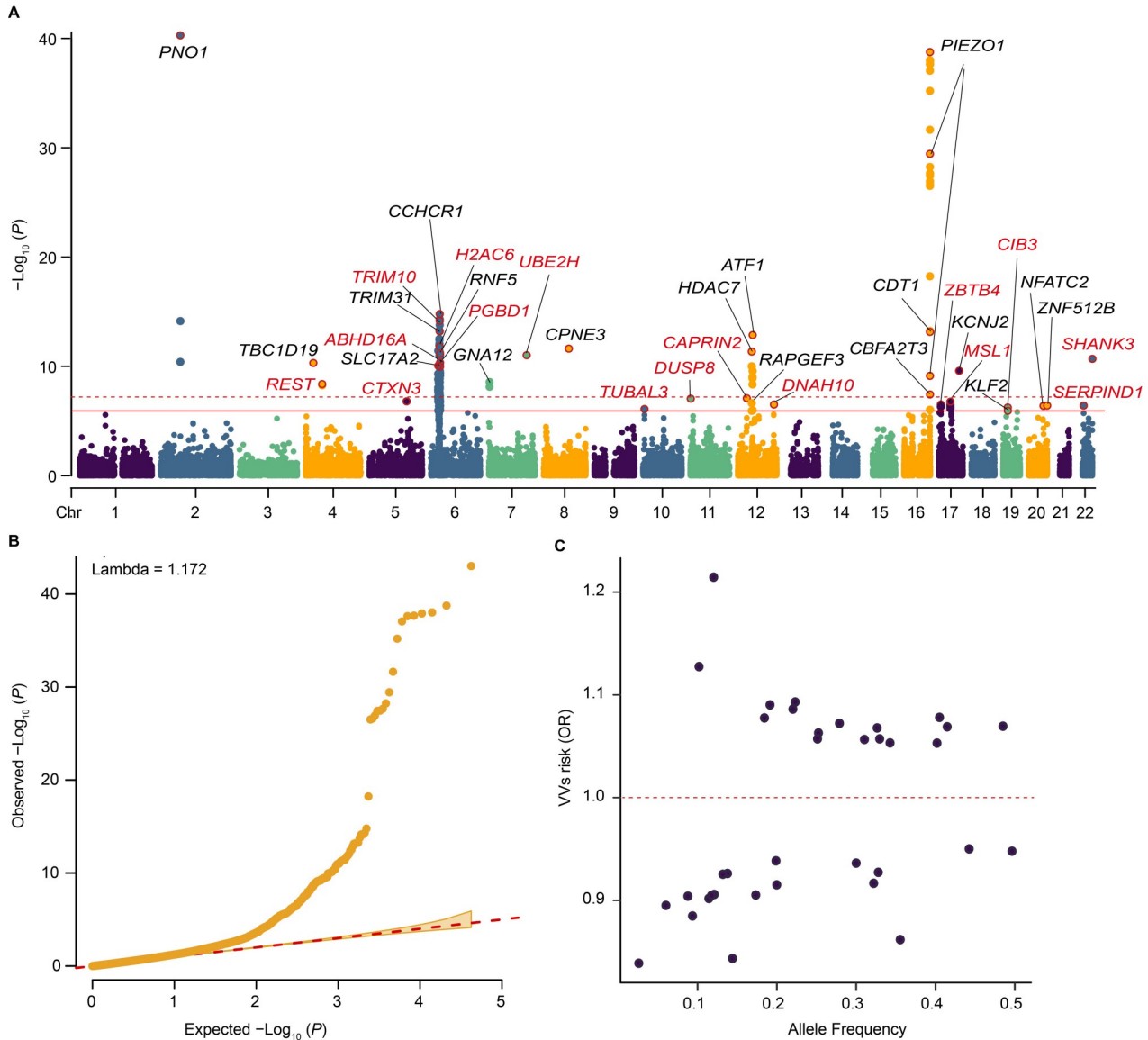

**Fig 1. Summary of single-variant associations with VV among the unrelated Caucasians. (A)** Manhattan plot of single-variant associations for common variants and VV. The red solid line indicates the significance threshold ($P = 1.19 \times 10^{-6}$), and the red dashed line indicates the genome-wide significance threshold ($P = 5.0 \times 10^{-8}$). The gene name corresponds to the gene closest to the variant. Black fonts indicate previously identified genes, while red fonts indicate newly identified genes. **(B)** The quantile-quantile (Q-Q) plot for single-variant analysis. **(C)** The plot of effect size (odds ratio) versus effect allele frequency of identified independent common variants. Abbreviation: OR, odds ratio.

The significance level of the genetic variant associations between these significant genes and VV was enhanced by the inclusion of missense variants. In the LOF + missense model with MAF < 1%, these 3 genes were significantly associated with an elevated risk of VV (*PIEZO1*, odds ratio (OR) = 1.010, 95% confidence interval (CI) = 1.008–1.013, $P = 1.15 \times 10^{-18}$; *ECE1*, OR = 1.028, 95% CI = 1.016–1.039, $P = 4.89 \times 10^{-7}$; *FBLN7*, OR = 1.012, 95% CI = 1.007–1.017, $P = 1.93 \times 10^{-6}$) (**S7 Table**). The gene which had the strongest correlation with VV was *PIEZO1* in the LOF model with MAF < $10^{-3}$ (*PIEZO1*, OR = 1.037, 95% CI = 1.029–1.046, $P = 5.01 \times 10^{-17}$). The associations of *PIEZO1* and *FBLN7* with VV have been reported in previous genetic studies [14,19]. The significant difference between the LOF + missense model

**Table 1. Independent common variants significantly associated with VV in the single-variant association analysis.**

| rsID | CHR | POS | A0 | A1 | OR | 95% CI | P | Gene | Exonic function | FinnGen VV P |
|---|---|---|---|---|---|---|---|---|---|---|
| rs2044693 | 2 | 68157965 | G | A | 0.86 | 0.84–0.88 | $9.81\times10^{-44}$ | PNO1 | Synonymous | $6.00\times10^{-24}$ |
| rs2227901 | 4 | 56932023 | G | A | 1.08 | 1.05–1.11 | $4.23\times10^{-9}$ | **REST** | Synonymous | $1.94\times10^{-6}$ |
| | 4 | 26584259 | C | T | 1.07 | 1.05–1.09 | $4.97\times10^{-11}$ | TBC1D19 | Synonymous | / |
| rs248709 | 5 | 127657571 | T | A | 1.06 | 1.03–1.08 | $1.59\times10^{-7}$ | **CTXN3** | Synonymous | $6.28\times10^{-3}$ |
| rs34525648 | 6 | 25914625 | G | A | 0.91 | 0.88–0.93 | $4.87\times10^{-10}$ | SLC17A2 | Nonsynonymous | $7.02\times10^{-7}$ |
| | 6 | 26124406 | C | T | 1.09 | 1.06–1.11 | $3.81\times10^{-12}$ | **H2AC6** | Synonymous | / |
| rs33932084 | 6 | 28301047 | A | G | 0.90 | 0.88–0.93 | $6.28\times10^{-10}$ | **PGBD1** | Nonsynonymous | $1.08\times10^{-7}$ |
| rs2239529 | 6 | 30110553 | C | T | 0.91 | 0.88–0.93 | $3.07\times10^{-13}$ | TRIM31 | Synonymous | $5.61\times10^{-4}$ |
| rs3094134 | 6 | 30154377 | C | T | 1.13 | 1.09–1.16 | $5.29\times10^{-14}$ | **TRIM10** | Synonymous | $4.20\times10^{-4}$ |
| rs1576 | 6 | 31142614 | G | C | 0.92 | 0.90–0.94 | $1.65\times10^{-15}$ | CCHCR1 | Nonsynonymous | $4.23\times10^{-6}$ |
| rs1475865 | 6 | 31689636 | C | T | 1.07 | 1.05–1.10 | $2.73\times10^{-10}$ | **ABHD16A** | Synonymous | $1.08\times10^{-4}$ |
| rs3130349 | 6 | 32179919 | G | A | 0.92 | 0.89–0.94 | $7.02\times10^{-12}$ | RNF5 | Synonymous | $1.47\times10^{-4}$ |
| rs12539800 | 7 | 129839298 | T | C | 0.88 | 0.85–0.92 | $9.67\times10^{-12}$ | **UBE2H** | Synonymous | $2.92\times10^{-8}$ |
| rs798488 | 7 | 2762888 | T | C | 0.94 | 0.92–0.96 | $2.75\times10^{-9}$ | GNA12 | Start loss | $6.84\times10^{-3}$ |
| rs2304789 | 8 | 86554965 | C | T | 0.93 | 0.91–0.95 | $2.42\times10^{-12}$ | CPNE3 | Nonsynonymous | $2.99\times10^{-9}$ |
| rs7097775 | 10 | 5393955 | A | G | 1.05 | 1.03–1.08 | $8.09\times10^{-7}$ | **TUBAL3** | Synonymous | $3.46\times10^{-4}$ |
| rs7129499 | 11 | 1556860 | G | A | 0.95 | 0.93–0.97 | $9.14\times10^{-8}$ | **DUSP8** | Synonymous | $9.48\times10^{-7}$ |
| rs4930721 | 12 | 123933342 | C | T | 1.06 | 1.03–1.08 | $3.14\times10^{-7}$ | **DNAH10** | Synonymous | $7.74\times10^{-5}$ |
| rs12146709 | 12 | 30735067 | T | C | 1.06 | 1.04–1.09 | $8.44\times10^{-8}$ | **CAPRIN2** | Nonsynonymous | $4.72\times10^{-3}$ |
| rs12422983 | 12 | 47748853 | C | T | 0.93 | 0.90–0.95 | $2.32\times10^{-7}$ | RAPGEF3 | Nonsynonymous | $4.99\times10^{-4}$ |
| rs7306788 | 12 | 47784123 | C | T | 1.09 | 1.06–1.12 | $4.46\times10^{-12}$ | HDAC7 | Synonymous | $8.05\times10^{-15}$ |
| rs1129406 | 12 | 50809588 | C | T | 1.08 | 1.06–1.10 | $1.37\times10^{-13}$ | ATF1 | Synonymous | $3.91\times10^{-18}$ |
| rs8043924 | 16 | 88721296 | C | G | 1.21 | 1.18–1.25 | $1.75\times10^{-39}$ | PIEZO1 | Synonymous | $4.56\times10^{-37}$ |
| rs34908386 | 16 | 88734505 | C | T | 0.84 | 0.82–0.87 | $5.74\times10^{-29}$ | PIEZO1 | Synonymous | $4.28\times10^{-28}$ |
| rs6500493 | 16 | 88737574 | G | C | 1.07 | 1.05–1.09 | $7.48\times10^{-10}$ | PIEZO1 | Synonymous | $2.52\times10^{-6}$ |
| rs510862 | 16 | 88806103 | C | T | 1.09 | 1.07–1.12 | $5.79\times10^{-14}$ | CDT1 | Synonymous | $4.93\times10^{-14}$ |
| rs61734177 | 16 | 88898149 | C | G | 0.90 | 0.87–0.94 | $3.77\times10^{-8}$ | CBFA2T3 | Nonsynonymous | $6.78\times10^{-4}$ |
| rs143879261 | 17 | 40129440 | C | A | 0.84 | 0.78–0.90 | $1.71\times10^{-7}$ | **MSL1** | Synonymous | 0.03 |
| rs173135 | 17 | 70176185 | C | T | 0.90 | 0.87–0.93 | $2.58\times10^{-10}$ | KCNJ2 | Synonymous | $1.18\times10^{-5}$ |
| rs34914463 | 17 | 7463300 | T | C | 0.93 | 0.90–0.95 | $3.10\times10^{-7}$ | **ZBTB4** | Nonsynonymous | $2.19\times10^{-3}$ |
| rs6512087 | 19 | 16164844 | T | C | 0.94 | 0.91–0.96 | $5.86\times10^{-7}$ | **CIB3** | Synonymous | 0.01 |
| rs3745318 | 19 | 16325451 | C | T | 1.06 | 1.03–1.08 | $1.18\times10^{-6}$ | KLF2 | Synonymous | $4.16\times10^{-10}$ |
| rs3746420 | 20 | 51524088 | G | C | 0.89 | 0.86–0.94 | $4.15\times10^{-7}$ | NFATC2 | Synonymous | $1.13\times10^{-8}$ |
| rs817329 | 20 | 63966341 | G | T | 1.05 | 1.03–1.07 | $3.92\times10^{-7}$ | ZNF512B | Synonymous | $1.02\times10^{-9}$ |
| rs4675 | 22 | 20787012 | T | C | 0.95 | 0.93–0.97 | $3.94\times10^{-7}$ | **SERPIND1** | Synonymous | $7.97\times10^{-3}$ |
| rs9616915 | 22 | 50679152 | C | T | 1.07 | 1.05–1.09 | $2.07\times10^{-11}$ | **SHANK3** | Synonymous | $8.60\times10^{-6}$ |

Genes highlighted in bold are not previously reported. rsID was obtained from FinnGen. The "/" represents the variants not tested in FinnGen.

Abbreviations: VV, varicose veins; CHR, chromosome; POS, position; OR, odds ratio; CI, confidence interval.

($P = 4.89 \times 10^{-7}$) and the LOF-only model ($P = 0.02$) for *ECE1*, a novel identified VV-related gene, suggested its association with VV was primarily driven by the missense mutation (**Table 2**). The associations of the three genes we identified in collapsing analysis with VV were further successfully validated in the VV genomic data from FinnGen (*P* ranged from $2.39 \times 10^{-59}$ to $4.93 \times 10^{-5}$, **Table 2**).

Considering the high prevalence and incidence of VV in the general population, we calculated the carrier frequencies and disease prevalence rates of the putative pathogenic variants.

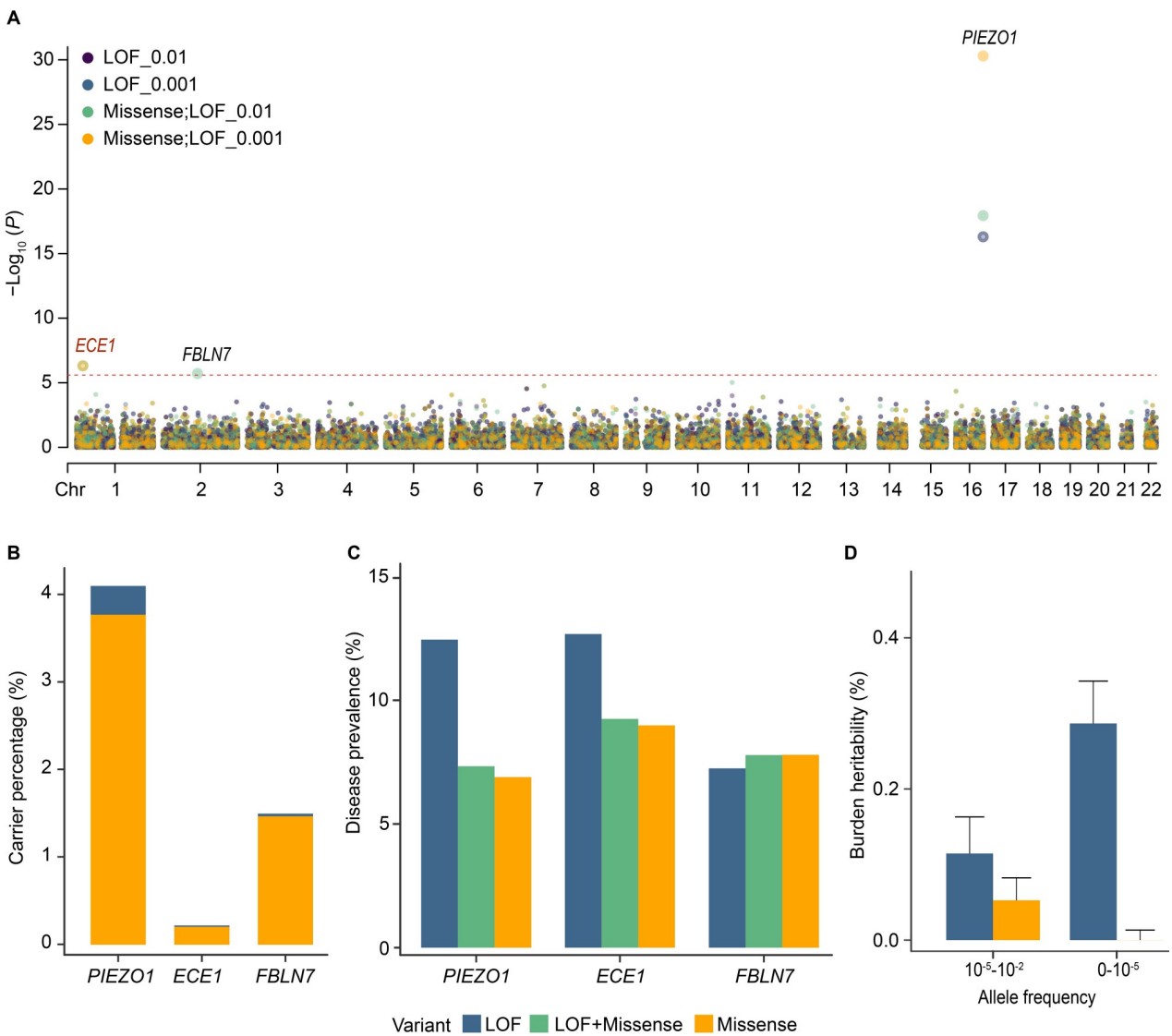

**Fig 2. Summary of rare genetic associations with VV among the unrelated Caucasians. (A)** Manhattan plot of the gene-based collapsing analysis. The red dashed line indicates the significance threshold ($P = 2.51 \times 10^{-6}$). We only showed the results of the collapsing test with four models including two MAF cutoffs (0.01 and 0.001) and two variant annotation groups (LOF and LOF + missense), and the other results are shown in **S6 Table**. All models were adjusted for age, sex, and the top 10 PCs. **(B)** Bar chart showing carrier frequencies for rare LOF and missense variants for three identified rare variant genes. **(C)** Disease prevalence of LOF and missense variants in VV-related rare variant genes. **(D)** Heritability of the burden of genetic variants in VV. Abbreviation: LOF, loss-of-function.

*PIEZO1* had the highest carrier frequency of missense variants (3.77%, the probability of being loss-of-function intolerant (pLI) = 0.54), followed by *FBLN7* (1.47%, pLI = 0) and *ECE1* (0.20%, pLI = 1) (**Fig 2B** and **S8** and **S9 Tables**). The high pLI of *ECE1* underscores its pronounced intolerance to LOF variants, consistent with our findings of lower carrier frequency (**S4 Fig**). These variant carriers showed generally modest disease prevalence rates (< 15%), and the disease prevalence rates of missense variants were lower than those of LOF variants in carriers (**Fig 2C**).

**Table 2. Significant associations with VV in the gene-level collapsing analysis.**

| Gene | Group | MAF | Number of rare variants | OR | 95% CI | P | FinnGen VV P |
|------|-------|-----|------------------------|-----|--------|---|-------------|
| *PIEZO1* | LOF | 0.01 | 204 | 1.037 | 1.029–1.046 | $5.01 \times 10^{-17}$ | $2.39 \times 10^{-59}$ |
| | LOF + Missense | 0.01 | 735 | 1.010 | 1.008–1.013 | $1.15 \times 10^{-18}$ | |
| | LOF | 0.001 | 204 | 1.037 | 1.029–1.046 | $5.01 \times 10^{-17}$ | |
| | LOF + Missense | 0.001 | 730 | 1.021 | 1.017–1.024 | $5.05 \times 10^{-31}$ | |
| | LOF | $1 \times 10^{-4}$ | 203 | 1.056 | 1.045–1.067 | $3.24 \times 10^{-24}$ | |
| | LOF + Missense | $1 \times 10^{-4}$ | 716 | 1.028 | 1.023–1.033 | $6.70 \times 10^{-28}$ | |
| | LOF | $1 \times 10^{-5}$ | 183 | 1.049 | 1.033–1.066 | $2.20 \times 10^{-9}$ | |
| | LOF + Missense | $1 \times 10^{-5}$ | 578 | 1.037 | 1.028–1.046 | $2.67 \times 10^{-16}$ | |
| **ECE1** | LOF | 0.01 | 30 | 1.051 | 1.007–1.096 | 0.02 | $4.93 \times 10^{-5}$ |
| | LOF + Missense | 0.01 | 189 | 1.028 | 1.016–1.039 | $4.89 \times 10^{-7}$ | |
| | LOF | 0.001 | 30 | 1.051 | 1.007–1.096 | 0.02 | |
| | LOF + Missense | 0.001 | 189 | 1.028 | 1.016–1.039 | $4.89 \times 10^{-7}$ | |
| | LOF | $1 \times 10^{-4}$ | 30 | 1.051 | 1.007–1.096 | 0.02 | |
| | LOF + Missense | $1 \times 10^{-4}$ | 186 | 1.034 | 1.020–1.049 | $1.12 \times 10^{-6}$ | |
| | LOF | $1 \times 10^{-5}$ | 30 | 1.051 | 1.007–1.096 | 0.02 | |
| | LOF + Missense | $1 \times 10^{-5}$ | 174 | 1.041 | 1.024–1.059 | $1.37 \times 10^{-6}$ | |
| *FBLN7* | LOF | 0.01 | 29 | 1.007 | 0.978–1.036 | 0.60 | $7.25 \times 10^{-10}$ |
| | LOF + Missense | 0.01 | 136 | 1.012 | 1.007–1.017 | $1.93 \times 10^{-6}$ | |
| | LOF | 0.001 | 29 | 1.007 | 0.978–1.036 | 0.60 | |
| | LOF + Missense | 0.001 | 135 | 1.011 | 1.003–1.019 | 0.01 | |
| | LOF | $1 \times 10^{-4}$ | 29 | 1.007 | 0.978–1.036 | 0.60 | |
| | LOF + Missense | $1 \times 10^{-4}$ | 128 | 1.010 | 0.998–1.022 | 0.20 | |
| | LOF | $1 \times 10^{-5}$ | 25 | 0.986 | 0.941–1.034 | 0.56 | |
| | LOF + Missense | $1 \times 10^{-5}$ | 102 | 1.004 | 0.982–1.027 | 0.70 | |

Genes highlighted in bold are not previously reported. Number of rare variants: number of markers with MAF < 0.01.

Abbreviations: VV, varicose veins; MAF, minor allele frequency; OR, odds ratio; CI, confidence interval; LOF, loss of function.

## Burden heritability of VV

Then we calculated the gene-wise burden heritability of VV using the method developed by Weiner et al. [22]. In this analysis, ultra-rare variants were defined as MAF < $1 \times 10^{-5}$ and rare variants were defined as having a MAF between $1 \times 10^{-5}$ and $1 \times 10^{-2}$. The burden heritability of rare variants focuses on variants that predict the most serious functional consequences. Among these, ultra-rare coding LOF variants explained 0.287% (se = 0.056%) of phenotypic variance, more than rare LOF variants explained. But rare missense variants accounted for more burden heritability than ultra-rare (rare, 0.053%; ultra-rare, -7.63 × $10^{-7}$) (**Fig 2D and S10 Table**).

## Robustness of the association of rare variant genes with VV

We performed leave-one-variant-out (LOVO) analyses to assess the robustness of associations identified in gene-based collapsing analysis and to find the variants that affect each VV-related rare variant gene (**S11 Table**). After the removal of the variant chr1:21227980:A:G, the association between *ECE1* and VV became non-significant ($P = 3.81 \times 10^{-6}$). Similarly, the highest LOVO P value attained for the association was $P = 9.90 \times 10^{-3}$ for *FBLN7* and VV after removing chr2:112165024:C:G. Importantly, the association of VV and *PIEZO1* remained robust in the LOVO analysis (**S5 Fig**). This suggested that the significant associations of VV with *ECE1*

and *FBLN7* were strongly influenced by single rare variants, whereas the association with *PIEZO1* were more influenced by a combined effect of multiple rare variants. We further assessed whether the significant rare variants were independent of nearby common variants by a conditional analysis. The results showed that the nearby common variants did not affect the effect size and significance of these three rare variant genes (**S12 Table**).

## Sensitivity analysis of the effects of sex on VV

To explore whether the effects of genes on VV differ by sex, we re-performed single-variant analysis and collapsing analysis after stratifying participants by sex. These analyses included 162,210 males and 188,560 females. After clumping, we identified 5 and 21 independent common variants significantly related to VV in males and females, respectively, with four variants mapping to novel genes only in females (*ARHGEF26*, *BTN1A1*, *H1-5*, *TRIM27*, **S13 Table**). The *PIEZO1* gene was strongly associated with VV, with different significance levels across sexes (male, $P = 4.66 \times 10^{-14}$; female, $P = 6.81 \times 10^{-21}$). The significant association between *FBLN7* and VV was found only in females ($P = 1.12 \times 10^{-6}$). Notably, we additionally identified a significant association between a novel gene (*METTL21A*, $P = 1.38 \times 10^{-6}$) and VV in females in the model of LOF variants with MAF $< 1 \times 10^{-4}$ (**S14 Table**).

## Enrichment analysis of VV genes and their expression

We conducted a pathway enrichment analysis using the Gene Ontology (GO) Consortium [23,24] for all 36 genes identified in the study as being significantly associated with VV. There were 12 GO terms significantly enriched for these genes (false discovery rate (FDR) correction, corrected α = 0.05). It was found that several VV genes (including *UBE2H*, *TRIM31*, and *RNF5*) were significantly enriched in protein K48-linked ubiquitination ($P = 7.20 \times 10^{-3}$) and 30 genes were enriched in protein binding ($P = 2.79 \times 10^{-2}$) (**Fig 3A and S15 Table**). In females, GO enrichment analysis showed that the identified VV-related genes were significantly enriched in protein ubiquitination ($P = 1.47 \times 10^{-3}$) and most genes were enriched in protein binding ($P = 1.48 \times 10^{-2}$) (**Fig 3B and S16 Table**). However, the VV genes identified in males were not successfully enriched for the same biological processes.

In addition, we explored the expression of VV-related independent common and rare variant genes in 30 general tissues in FUMA GTEx. We discovered that several genes were highly expressed in adipose tissue, including *PIEZO1* and the novel identified *ECE1* (**Fig 3C and S17 Table**). Next, single-cell expression data from human adipose tissue showed that *PIEZO1* and *ECE1* were most strongly expressed in the endothelial cell subpopulation, and higher expression of *ECE1* was also detected in fibroblasts and neutrophils (**Fig 3D, 3E and 3F**).

## Phenotypic associations between identified VV genes and multiple traits

To further explore the association between VV-related genes and other phenotypes, we performed phenome-wide association analysis (Phe-WAS) for multiple quantitative traits and binary phenotypes, including biochemical indicators, cardiovascular disease, body size measures, inflammatory indicators, and pulmonary function (**S18 Table**). As expected, the majority of 53 traits showed significant associations with the identified VV-related genes ($P < 2.77 \times 10^{-5}$), especially common variant genes such as *RNF5* (27/53), *PGBD1* (24/53), *TRIM31* (24/53), and *GNA12* (14/53) (**Fig 4**). Furthermore, among these significant genotype-phenotype associations of independent common variant genes, 168 associations were negative and 113 were positive (**S19 Table**). Among three rare variant genes, *PIEZO1* was significantly associated with a variety of curated phenotypes, such as HbA1c ($P = 1.02 \times 10^{-75}$), standing height ($P = 5.41 \times 10^{-23}$), sitting height ($P = 7.23 \times 10^{-16}$), and forced expiratory volume in 1

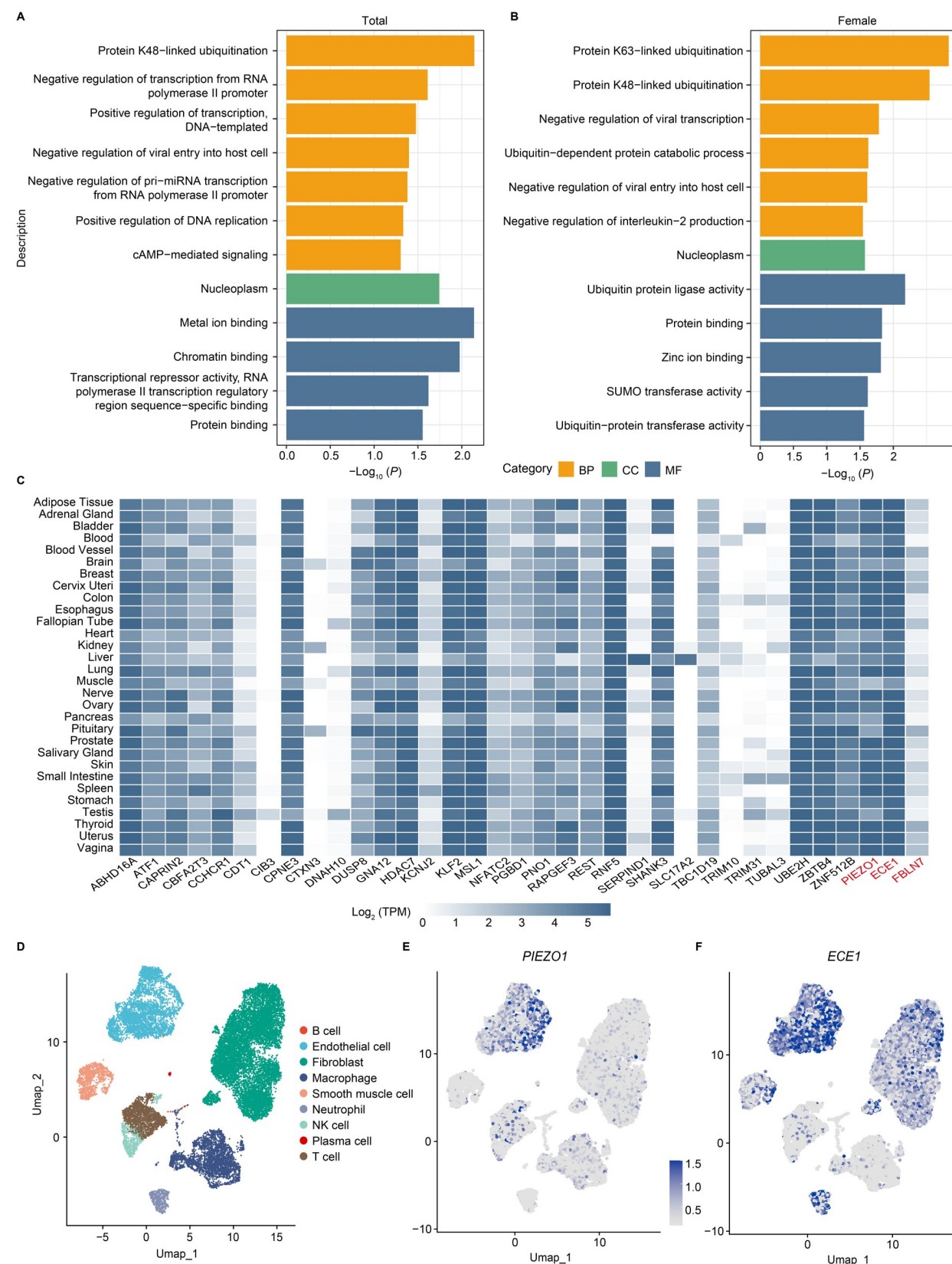

**Fig 3. Enrichment and expression analysis of significant effect associations in VV.** (A) Results of the functional enrichment in biological pathway databases (*P* < 0.05). (B) Results of the functional enrichment in biological pathway databases in females. Only the top 12 significant pathways (*P* < 0.05) are shown, and the complete results are shown in **S16 Table**. (C) The heat map shows the tissue enrichment results of each VV-related gene. The *H2AC6* gene was not recognized in FUMA Ensembl ID. (D) Single-cell RNA sequencing analysis results of adipose tissue. (E, F) Cell-specific expression of *PIEZO1* and *ECE1* in adipose tissue. Abbreviation: BP, biological process; CC, cellular component; MF, molecular function; TPM, transcripts per million; Umap, Uniform manifold approximation and projection.

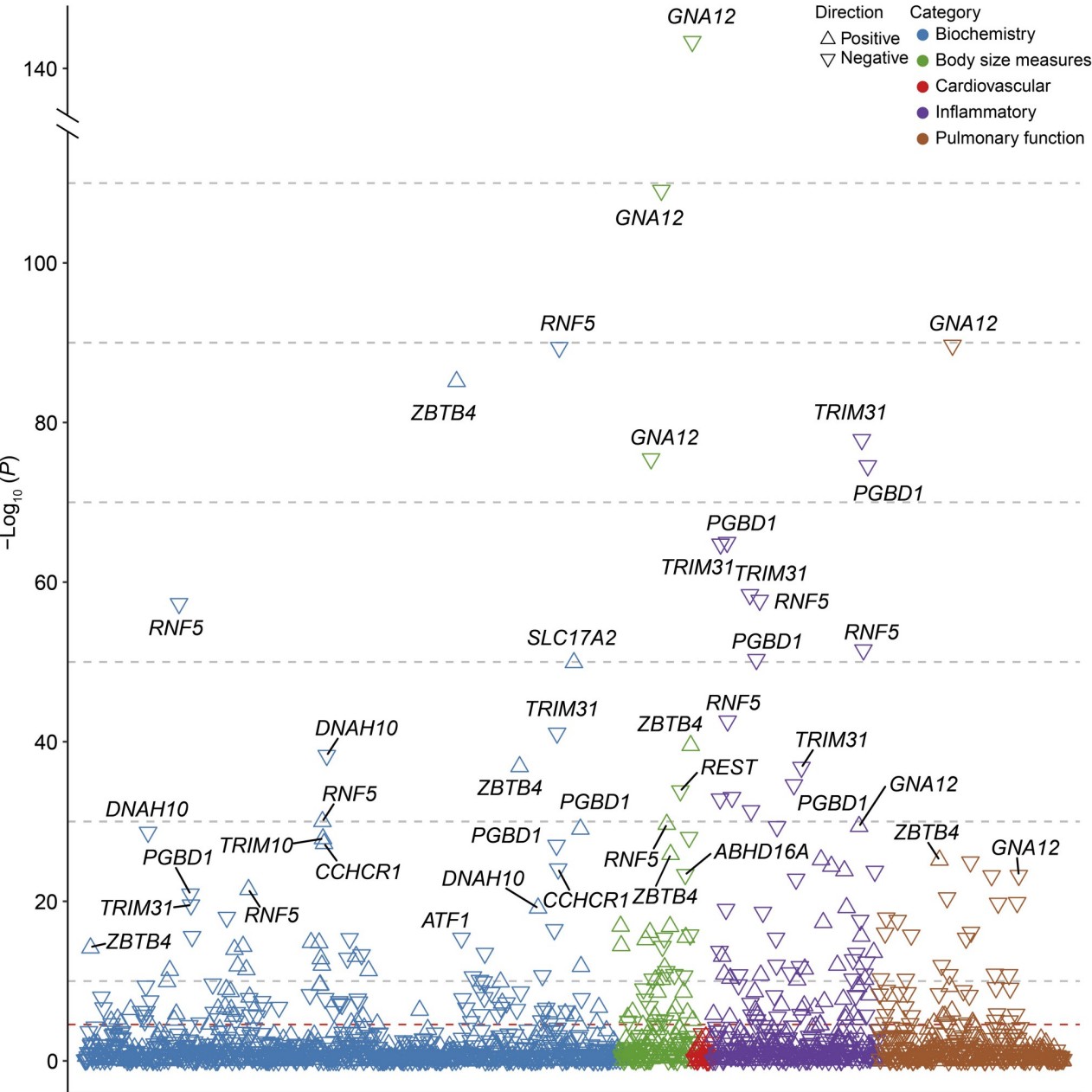

**Fig 4. Phenotypic associations of the independent common variant genes associated with VV.** The *P*-values shown are two-sided. Multiple-trait Manhattan plot representing the association results between common variant genes and 53 phenotypes. The X-axis shows various phenotypes listed in the **S19 Table**. The dashed line represents the Bonferroni-corrected significance threshold for 53 traits and 34 genes ($P = 2.77 \times 10^{-5}$).

second (FEV1 pred, $P = 1.07 \times 10^{-12}$), and *FBLN7* was mainly linked to height-related traits and pulmonary function indices (**Fig 5** and **S20 Table**).

In addition, we tried to find genes with similar continuous trait fingerprints to these rare variant genes on the Gene-SCOUT website [25]. *SCUBE3* and *IGF2BP2* were found to be the most similar to the *PIEZO1*, respectively (**Fig 6A**). The module of gene signatures showed that

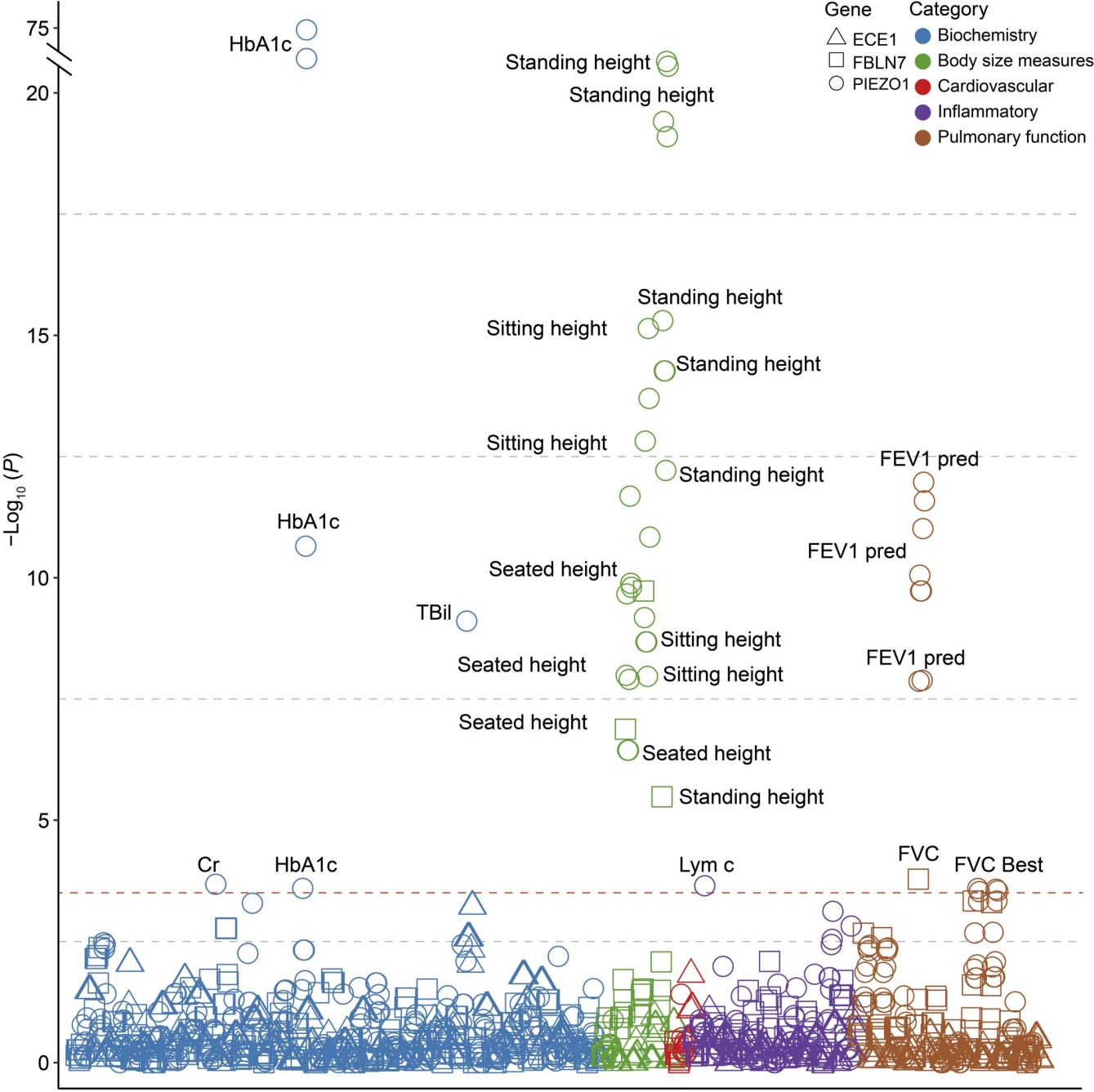

**Fig 5. Phenotypic associations of the rare variant genes associated with VV.** The *P*-values shown are two-sided. Multiple-trait Manhattan plot representing the association results between 3 rare variant genes and 53 phenotypes. X-axis shows various phenotypes listed in the **S20 Table**. The dashed line represents Bonferroni-corrected significance for 53 traits and 3 genes ($P = 3.14 \times 10^{-4}$). Abbreviations: HbA1c, glycated hemoglobin; FEV1 pred, forced expiratory volume in 1-second, predicted; FVC, forced vital capacity; TBil, total bilirubin; Lym c, lymphocyte count; FVC Best, forced vital capacity, best measure.

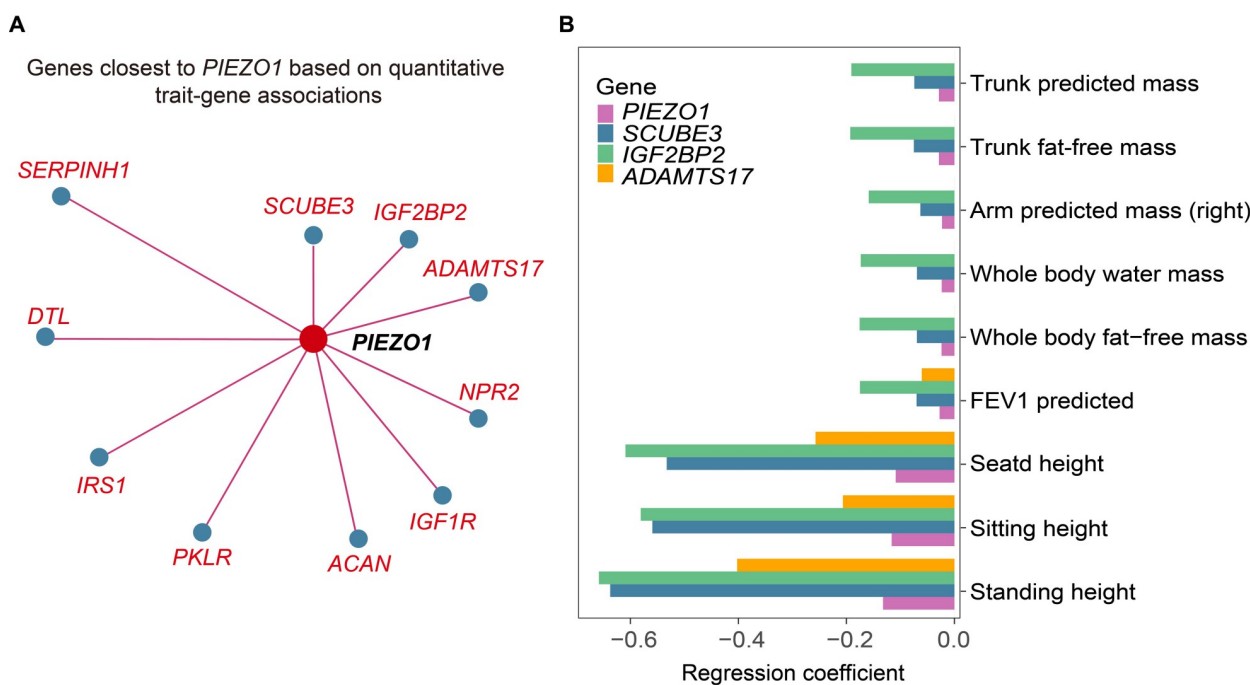

**Fig 6. Quantitative traits and disease characteristics of *PIEZO1*. (A)** Genes with the most similar quantitative trait characteristics to *PIEZO1* in UK Biobank from Gene-SCOUT. **(B)** Signatures of *PIEZO1* and its similar genes (similarity to *PIEZO1* sorted from top to bottom).

both these similar genes and *PIEZO1* were negatively associated with height related traits (**Fig 6B**). Recent evidence using machine learning methods [15] has identified that height is a risk factor for VV. Our findings further validated the effect of height on the risk of developing VV.

## Discussion

Previous GWAS [14,15,26] have identified multiple risk loci and pathways involved in the pathophysiology of VV, improving the understanding of the polygenic architecture of the disease. The WES data from the UKB could provide a new direction for exploring the genetic mechanisms of VV [27]. VV-related genes and genetic variants identified in population-based WES studies were more abundant and reliable. It is perhaps that whole-exome data are usually obtained directly from sequencing rather than by imputation. WES typically focuses on coding regions of the genome and is more cost-effective and widely available than whole-genome sequencing [28]. Compared to non-coding variants, WES can directly assess variants that alter protein sequence, making it easier to explain their functional consequences. It also provides a clearer pathway to deeper mechanistic insights, and can potentially be useful for therapeutic target discovery [29–32] and precision medicine [33,34].

Recently, several studies of human gene-phenotype association for rare coding variants have identified the association between *PIEZO1* and VV by using exome sequencing data from UKB participants [16,20]. Compared to the studies by Van Hout et al.[16] and Wang et al.[20], our study has a larger sample size and more validation of the findings (replication in the Finn-Gen cohort) and exploration of biological function (e.g. functional enrichment analysis, tissue expression analysis, and single-cell RNA sequencing analysis). In addition to the replication of 19 known VV-related genes from previous studies, our study further identified 17 novel genes. Most of these genes were successfully validated in FinnGen. Tissue analysis and single-cell gene expression analysis emphasized the importance of adipose tissue in the association

between these identified genes and VV risk. Our Phe-WAS revealed the strong associations of these genes with body size measures, biological indicators, and inflammatory markers.

We identified a total of 36 VV-associated genes in our large-scale exome sequencing study. First, the exome-wide single-variant analysis identified 36 VV-related independent common variants mapping to 34 genes, half of which have been previously established [14,15,26,35,36]. The significant effects found in the single-variant analysis were the effect of *PNO1* on chromosome 2 and that of *PIEZO1* on chromosome 16. *PNO1* plays an important role in both proteasome and ribosome biogenesis [37], and *PIEZO1* is required for vascular development and function [38]. *PNO1* is predominantly expressed in the liver and spleen, and slightly in the thymus, testis, and ovary, but not in the heart or brain [39]. In addition, it has been suggested that *PNO1* may be involved in the expression of calcineurin phosphatase, a key signaling component in angiogenesis, and calcineurin phosphatase activity is essential for vascular development [15,40]. As the most significant novel gene identified in the single-variant analysis, *TRIM10* was thought to be potentially involved in the body's immune response [41]. Phenotypic association analysis revealed that *TRIM10* had associations with several inflammatory markers, such as platelets, monocytes, and neutrophils. As a protein-coding gene, *UBE2H* encodes an E2 ubiquitin-conjugating enzyme family protein, which is involved not only in ubiquitination but also in cytoskeletal regulation and calcium signaling [42]. Our study also indicated that *UBE2H* is involved in protein binding and protein K48-linked ubiquitination by functional enrichment of the gene.

The other findings in this study were the associations of VV with three rare variant genes (*PIEZO1*, *ECE1*, and *FBLN7*) identified by the gene-based collapsing analysis. Both single-variant analysis and gene-based collapsing analysis revealed an association between VV and *PIEZO1*, supporting the previously reported hypothesis that common and rare variants may involve overlapping genes [43,44]. A study by Backman et al. [43] found that missense variants in *PIEZO1* might have a gain-of-function effect and reduce VV risk, which offers the possibility of treating VV by pharmacological interventions. *PIEZO1* is a mechanically activated ion channel that plays a decisive role in vascular architecture [38,45]. Its integral or endothelial-specific disruption can severely damage the growing vascular system and even lead to embryonic death.

As a newly discovered VV-related gene, *ECE1* is an estrogen-regulatory gene in mesenteric arteries that could catalyze the conversion of pro-endothelin into endothelin-1, a potent vasoconstrictor [46,47]. A mouse model showed that estrogen receptor stimulation suppressed *ECE1* expression [48]. In addition, previous studies have found an increase in *ECE1* in pathological conditions, for instance, hypertension, coronary atherosclerosis, and vascular injury models [49–52]. Our expression analysis of *ECE1* revealed its high expression in adipose tissue, especially in endothelial cells, and a positive association between obesity and VV risk has been demonstrated before [15,26]. It was well known that the development of VV was associated with a variety of vascular factors, such as changes in hemodynamics, endothelial cell activation, inflammation, and hypoxia [53–55]. As a cell adhesion molecule associated with diseases and developmental abnormalities, *FBLN7* plays an essential role in specialized tissues such as the placenta, blood vessels, and cartilage [56–58]. *FBLN7* protein fragments have been found to regulate endothelial cells and leukocyte function [59,60]. With anti-angiogenic function [57], they can be used to treat inflammatory diseases by modulating immune cell activity [61].

Moreover, although the carrier frequencies of LOF variants in the population were low, LOF mutations confer greater disease prevalence and genetic susceptibility to VV. As for the burden heritability, ultra-rare LOF variants had the highest heritability, followed by rare LOF variants and rare missense variants, which was in line with the previous finding by Weiner et al. [22] that LOF variants explained the majority of burden heritability.

We evaluated the robustness of the three genes identified in the collapsing analysis. The rare variant association results remained stable after adjusting for nearby common variants, probably due to our strict variant filter. Our sensitivity analysis revealed a novel gene in females (*METTL21A*), which prompted us to speculate whether the association between VV and sex might be influenced by genetic factors. Moreover, the current study revealed an overlap between VV-related genes identified by single-variant analyses and those identified by burden tests, suggesting the robustness of the association between overlapping genes and VV risk. A previous genetic study showed that vascular diseases were tightly linked to anthropometric indices [36]. The Phe-WAS of the identified genes showed VV genes were also associated with VV-related phenotypes, such as cardiovascular disease, body size measures, and pulmonary function indices. Phe-WAS can further explore genetic contributions to disease and identify the mechanisms of sharing across diseases [62].

There were some limitations in our study. First, the UKB was generally recruited from populations of European ancestry between the ages of 37 and 73, which makes our findings potentially inapplicable across all ethnic and age groups. Second, disease diagnoses are typically confirmed based on participant self-reports, ICD diagnosis codes, surgical records, and death registries. This may lead to the misclassification of disease phenotypes. Nevertheless, previous studies have been able to replicate genetic loci findings for common variants using the phenotype definitions in GWAS [14,15,26]. Third, we focused on the coding regions of the genome captured by WES, which may lead to the neglect of variants in non-coding regulatory regions. Due to the difficulty in obtaining exome sequencing data for VV from other databases, we validated our findings using summary statistics from FinnGen. The strength of our study is that it has the largest sample size of any VV-related whole-exome association study to date. Exome-wide analyses allow for more efficient targeting of variant genes that play an essential role in complex diseases compared to GWAS [63]. We defined the VV phenotype mainly based on the study of Ahmed et al.[14], which successfully replicated most of the VV-related risk loci identified in previous studies using this definition [15,64]. We additionally validated the results in FinnGen to determine the reproducibility of our findings.

In conclusion, large-scale exome-wide association studies are required for exploring the genetic associations of diseases. In this study, we identified many known genes and several novel effector genes significantly associated with VV. We found that LOF variants explained more burden heritability than missense variants. Furthermore, investigating genotype–phenotype associations would be critical for our insight into the underlying biological mechanisms of the disease, as well as for disease prevention and treatment. The study of genetic variants and diseases helps to develop strategies to target risk genes for prevention and treatment purposes.

## Methods

### Ethics statement

The UKB has received ethical approval from the National Health Service National Research Ethics Service and obtained written informed consent from all participants. This study was conducted under application number 19542.

### Study population

The UKB is a large-scale prospective cohort of nearly 500,000 individuals aged 37–73 recruited between 2006 and 2010 at 22 centers in England, Scotland, and Wales [65]. Each participant provided biological samples, phenotypic endpoints, registry information on cancer and death,

and health-related information, including biometric measures, lifestyle indicators, biomarkers in blood and urine, and imaging data [66].

In the discovery cohort, the VV cases were identified from the UKB using the following diagnostic, operative, and self-report codes (**S1 Table**): primary and/or secondary ICD-10 codes for varicose veins (I83), primary and/or secondary OPCS code for varicose vein surgery (L84-L88), self-reported operation code for varicose vein surgery (1479), self-reported non-cancer illness code for varicose veins (1494). Finally, 30,130 individuals who held at least one of these codes were classified as cases of VV.

## Sequencing and quality control

The Regeneron Genetics Center (RGC) performed whole-exome sequencing on all participants from the UKB. The complete sequencing protocols were described in detail in the previous study [16]. Samples were subjected to double exponential $75 \times 75$bp paired-end reads using the IDT xGen Exome Research Panel v1.0 on the NovaSeq 6000 platform to capture exomes with over 20X coverage at 95.2% of sites. We performed similar additional QC on the OQFE WES pVCF files in the GRCh38 human reference genome construct [67] provided by the UKB (https://biobank.ctsu.ox.ac.uk/showcase/label.cgi?id=170), which was similar to that in a previous study [68]. Due to the limited quality control (QC) and filtering of the published dataset, we used a high-quality dataset generated from an extensive genotype, variant, and sample-level pipeline for further analysis (**S1 Text**). We utilized Hail to perform the genotype quality control and split multi-allelic sites into bi-allelic sites. All calls with low genotype quality or critically low/high genotype depth were set to no calls. And then we excluded variants based on the call rate < 90%, Hardy-Weinberg $P$-value $< 1 \times 10^{-15}$, and in Ensembl low-complexity regions. We selected a set of high-quality independent autosomal variants for the relationship inference, which was selected as: MAF > 0.1%, missingness < 1%, Hardy–Weinberg equilibrium (HWE) $P$-value $> 1 \times 10^{-6}$ and two rounds of pruning using—indep-pairwise 200 100 0.1 and—indep-pairwise 200 100 0.05. KING [69] software was applied to calculate pairwise heterozygote concordance rates for each pair of samples and the kinship coefficient using high-quality variants. Sample-level QC involved removing samples with Ti/Tv, Het/Hom, SNV/indel, number of singletons exceeding the mean ± 8SD, and discrepancies between self-reported and genetically inferred sex, as well as duplicates and withdrawn consent. The kinship coefficient threshold of 0.0884 (the 2nd relatedness) was used to define a related sample. For pairs with kinship coefficients higher than 0.0884, we removed samples associated with multiple other individuals iteratively until none remained to maximize the sample size. Then we randomly removed one from the remaining pairs. Finally, we restricted the sample to Caucasians (field 22006) and used high-quality independent autosomal variants to calculate the PCs within the descent.

## Variant annotation

The preliminary analysis was performed by using SnpEff [70] to determine the most severe predicted consequences for specific variants in each gene transcript, with variants filtered to rare. We classified the variants annotated as stop gained, start/stop lost, splice donor/acceptor, or frameshift as LOF variants. A missense variant was considered damaging when it was predicted to be deleterious by SIFT (sorting intolerant from tolerant) [71], PolyPhen-2 HDIV, PolyPhen-2 HVAR [72], LRT (Likelihood Ratio Test)[73], and MutationTaster [74,75] consistently. The definitions of deleterious variants for each software are given in **S1 Text**.

## Single-variant association analysis

Single-variant association analysis of common exonic single nucleotide polymorphisms (SNPs) in VV was performed using SAIGE [76] software. The analysis was adjusted for age, sex, and the top 10 PCs, and the significance threshold was set at $1.19 \times 10^{-6}$ (Bonferroni correction for 42,123 exonic SNPs). Then, we clumped the results to remove possible associations due to strong linkage disequilibrium (LD) using the—clump function in PLINK v2. Significant SNPs within each risk locus were selected to identify the lead SNPs, which were defined as those with the lowest $P$ value when a strong LD ($r^2 > 0.01$) was observed for multiple SNPs within a 1000 kilobases (kb) window. Risk loci were identified as regions of ±1 Mb around lead SNPs.

## Gene-based exome-wide association analysis

We conducted gene-based collapsing analysis to define rare variant genes associated with VV. Rare variants were defined as MAF < 1% [77]. In our preliminary analysis, carriers of rare LOF variants and missense variants were decomposed into individual variables for each gene. We performed two functional category masks: LOF and LOF + missense variants, and subdivided each functional mask into four frequency masks: < 0.01, < 0.001, < 0.0001, and < 0.00001. It means that we constructed eight gene-burden models for each gene. Following adjustment for age, sex, and the top 10 PCs, gene-based collapsing analysis was conducted in SAIGE-GENE+ software [78] using a logistic mixed model approach with a saddle-point approximation. This approach is effective for the analysis of large sample data, and provides accurate $P$-values even when the case-control ratio is highly unbalanced [76]. In our collapsing analysis, we mainly reported the $P$ value of SKAT-O test [79], and the exome-wide significance threshold ($P = 2.51 \times 10^{-6}$) was determined by employing the Bonferroni correction (0.05/19,897). After identifying the VV-related genes, we further calculated the disease prevalence and carrier frequencies of rare variants for these genes.

## Burden heritability estimation

In this study, we estimated the heritability explained by the rare coding variants using burden heritability regression (BHR) [22]. BHR estimates burden heritability from the regression slope of the gene burden statistic on the burden score. We focused on estimating burden heritability for two functional categories (LOF and missense variants) and two allele frequency bins (MAF $< 1 \times 10^{-5}$, and MAF between $1 \times 10^{-5}$ and $1 \times 10^{-2}$). Ultra-rare coding variants were defined as MAF $< 1 \times 10^{-5}$ in this study. A single-variant association test was performed on the VV to obtain summary statistics of variant-level association. The "aggregate" mode was used to calculate the total burden heritability of disease across multiple frequency-function strata. Single-variant association summary statistics were used as input utilizing the baseline-BHR file provided by Weiner et al.[22]. These analyses were performed based on the R package bhr (v0.1.0).

## Replication of genes in the FinnGen cohort

We replicated the VV-related genes that we have identified using summary statistics from the FinnGen Consortium online results (version 8, 21). The FinnGen cohort recruited 342,499 individuals of Finnish descent with genotype and health register data, including 26,720 VV cases and 295,014 controls. The cohort collected and confirmed participants' diagnostic information through national healthcare registries and used the International Classification of Diseases (ICD) for documentation. Genetic association tests were performed using logistic mixed

models in SAIGE, and GWAS summary statistics for "Varicose veins" are publicly available online (see Data availability). To validate the findings of single-variant association analysis, we directly searched for our lead SNPs in summary statistics. For VV-related genes identified in gene-based collapsing analysis, we selected the variants with the strongest FinnGen VV correlation mapped to these genes. The significance threshold used for validation in the summary statistics was $P < 0.05$.

## Gene set enrichment and expression analyses

To identify terms and metabolic pathways with functional enrichment in differentially expressed genes, we performed the GO enrichment analysis using the Database for Annotation, Visualization, and Integrated Discovery (DAVID) [80]. The enrichment significance threshold was set at 0.05 (correction method = "FDR"). Normalized gene expression (transcripts per million, TPM) was obtained from the Genotype-Tissue Expression (GTEx) v8 for 30 general tissue types using the GENE2FUNC core program in Functional Mapping and Annotation of Genome-Wide Association Studies (FUMA GWAS) [81]. We used the average $\log_2$ (TPM) to compare expression levels between genes and tissue types. Data from each gene set was enriched for testing and multiple corrections (default for Benjamini-Hochberg). We leveraged the adipose tissue single-cell RNA sequencing data provided by the Tabula Sapiens Consortium to identify the expression levels of the genes in different cell types [82]. The Tabula Sapiens is a single-cell transcriptomic atlas of 24 human tissues and organs from 15 unique donors. The adipose tissues from two donors were obtained from surgery and immediately prepared from a fluorescence-activated cell sorting-sorted smart-seq2 platform or 10x Genomics 3' V3.1 droplet-based sequencing platform. The quality control, data integration, clustering, and cell type annotation procedures of single-cell RNA sequencing data were conducted, as described in the previous study [82]. The R package Seurat was used for downstream data analysis (i.e., normalization and data scaling) and visualization [83].

## Sensitivity and leave-one-variant-out analyses

As being female is one of the risk factors for VV, sensitivity analyses stratified by sex were performed to explore the sex heterogeneity of the genetic associations identified. There were 162,210 participants in the male group and 188,560 in the female group after stratification by sex. Enrichment analysis was then performed separately for the VV-related genes identified by sex. In addition, we assessed the robustness of the genetic results in the variant masks using LOVO analysis. After removing each variant from the significant models one at a time, we reran the association analysis for the variants included in the original mask. The variant that reached the maximum LOVO $P$ value after removal was regarded as the most important variant in each gene [68].

## Conditional analyses

We conducted multi-SNP-based conditional and joint association analyses (GCTA-COJO) using GCTA software [84,85] to identify conditionally independent variants from common variants associated with VV and to validate the results after clumping. The significance threshold was set at $P = 1.19 \times 10^{-6}$, and other parameters were left at their default settings.

We performed the conditional analysis to demonstrate that the significant rare variant signals were independent of nearby common variants. First, we ran association analyses for common variants (MAF $> 0.5\%$) within the ±500 kb genomic region of the identified genes. Then we used the—clump function (—clump-p1 $1 \times 10^{-5}$,—clump-r$^2$ 0.01) in PLINK [86] to aggregate and threshold the results, identifying independent index common variants. Finally, gene-

based collapsing analysis was rerun after adding the clumped common variant to the model as a covariate [68].

## Phenome-wide association analyses

To better explain the mechanisms of associations between risk genes and VV, we performed Phe-WAS. This analysis mainly focused on 53 phenotypes selected from UKB that may be associated with VV, including cardiovascular disease, 29 biochemical traits, 4 body size traits, 9 inflammatory traits, and 10 spirometry traits. Associations of 53 phenotypes with rare variant genes associated with VV were conducted using gene-level collapsing analysis ($P < 0.05/159$), whereas linear or logistic models were applied to independent common variants ($P < 0.05/1,802$). All models were adjusted for age, sex, and the top 10 PCs. To correct for multiple testing, the Bonferroni test was utilized.

## Gene-SCOUT

The Gene-SCOUT (Gene-Similarity from Continuous Traits, https://astrazeneca-cgr-publications.github.io/gene-scout/) [25] aims to find genes similar to specific genes and to construct a unique signature for each gene using associations derived from 450,000 exomes and 120,000 samples of metabolomic data sequenced from the UKB. This tool was used to identify the genes most similar to VV-related genes and to show significant associations of these genes with all quantitative traits.

## Supporting information

**S1 Fig. Age distribution of VV case and control groups.** Abbreviation: VV, varicose veins. (TIF)

**S2 Fig. Principal component plots of the VV group versus the non-VV group.** The x-axis and y-axis represent the values of the two components of PCA (PC1, PC2), and each point in the figure represents an individual. Abbreviation: VV, varicose veins; PC, principle components. (TIF)

**S3 Fig. The quantile-quantile (Q-Q) plots for gene-based collapsing analysis of VV.** The y-axis represents the observed–$\log_{10} P$ values across all tests, while the x-axis represents the expected under the null-hypothesis. $P$ values were obtained from the results of the gene-based collapse tests for varicose veins, using SAIGE-GENE+ software. In the gene-based collapse test, we applied two different maximum MAF cutoffs (0.01 and 0.001) and two different variant annotation groups (LOF and LOF + missense) to perform burden tests. All models were adjusted for age, sex, and top ten principal components. Abbreviation: VV, varicose veins; LOF, loss of function; λ, lambda. (TIF)

**S4 Fig. Inter-group differences in the probability of intolerance to loss of function between rare variant genes.** $P$-values for differences between groups were calculated by t-test. MAF, minor allele frequency, NS., no significance, pLI, the probability of intolerance to loss of function. (TIF)

**S5 Fig. Leave-one-variant-out (LOVO) analysis for rare variant genes associated with VV.** The x-axis indicates variants removed from the gene-based collapsed test, and the y-axis indicates the -$\log_{10} P$-value for associations without that variant $P$-values were derived from the

gene-based collapsed test using the SAIGE-GENE+ software, adjusted for age, sex, and the top 10 principal components.
(TIF)

**S1 Text. Supplementary Methods.**
(DOCX)

**S1 Table. Codes used for varicose veins case definition and the number of individuals with each of the diagnostic codes.**
(XLSX)

**S2 Table. Baseline characteristics of the study population.**
(XLSX)

**S3 Table. Significant results from single-variant associations analysis of VV ($P < 1.19 \times 10^{-6}$).**
(XLSX)

**S4 Table. Repeated validation of significant single-variant association results in the Finn-Gen.**
(XLSX)

**S5 Table. Independent common variants associated with VV after conditional and joint analysis.**
(XLSX)

**S6 Table. Collapsing analysis results for rare variants of VV.**
(XLSX)

**S7 Table. Significant effect values for the VV rare variant genes.**
(XLSX)

**S8 Table. Carrier status of rare variant genes with VV.**
(XLSX)

**S9 Table. The probability of being loss-of-function intolerant (pLI) of rare variant genes.**
(XLSX)

**S10 Table. Estimates of burden heritability across frequency bins and functional categories.**
(XLSX)

**S11 Table. Results from leave-one-out-variant (LOVO) analysis of rare variant genes associated with VV.**
(XLSX)

**S12 Table. Conditional analysis results on rare variant gene conditioning by nearby common variants.**
(XLSX)

**S13 Table. Sensitivity analysis results after stratified by sex (single-variant association analysis).**
(XLSX)

**S14 Table. Sensitivity analysis results after stratified by sex (collapsing analysis).**
(XLSX)

**S15 Table. GO enrichment analysis results of VV-associated genes.**
(XLSX)

**S16 Table. GO enrichment analysis results of VV-associated genes in females.**
(XLSX)

**S17 Table. GTEx tissue expression of VV-associated genes.**
(XLSX)

**S18 Table. Classification of phenotypes for phenome-wide association analysis.**
(XLSX)

**S19 Table. Phenome-wide association analysis results of VV-related independent common variant genes.**
(XLSX)

**S20 Table. Phenome-wide association analysis results of VV-related rare variant genes.**
(XLSX)

## Acknowledgments

We gratefully thank all the participants and professionals for collecting and preparing data in UKB and FinnGen study.

## Author Contributions

**Conceptualization:** Wei Cheng, Jin-Tai Yu.

**Data curation:** Liu Yang, Bang-Sheng Wu.

**Formal analysis:** Dan-Dan Zhang, Xiao-Yu He, Bang-Sheng Wu, Yan Fu, Wei-Shi Liu, Chen-Jie Fei, Ju-Jiao Kang.

**Methodology:** Xiao-Yu He, Liu Yang, Wei-Shi Liu, Ju-Jiao Kang.

**Project administration:** Dan-Dan Zhang.

**Software:** Liu Yang, Bang-Sheng Wu, Yan Fu, Chen-Jie Fei, Ju-Jiao Kang.

**Supervision:** Wei Cheng, Lan Tan, Jin-Tai Yu.

**Validation:** Xiao-Yu He.

**Visualization:** Dan-Dan Zhang, Wei-Shi Liu.

**Writing – original draft:** Dan-Dan Zhang, Xiao-Yu He.

**Writing – review & editing:** Yu Guo, Jian-Feng Feng, Lan Tan, Jin-Tai Yu.

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
