## [Decision Letter · Decision Letter 0]

11 Dec 2023

Dear Dr Yu,

Thank you very much for submitting your Research Article entitled 'Exome sequencing identifies novel genetic variants associated with varicose veins' to PLOS Genetics.

The manuscript was fully evaluated at the editorial level and by independent peer reviewers. The reviewers appreciated the attention to an important problem, but raised some substantial concerns about the current manuscript. Based on the reviews, we will not be able to accept this version of the manuscript, but we would be willing to review a much-revised version. We cannot, of course, promise publication at that time.

If you decide to revise the manuscript for further consideration at PLOS Genetics, please aim to resubmit within the next 60 days, unless it will take extra time to address the concerns of the reviewers, in which case we would appreciate an expected resubmission date by email to plosgenetics@plos.org.

We are sorry that we cannot be more positive about your manuscript at this stage. Please do not hesitate to contact us if you have any concerns or questions.

Yours sincerely,

Heather J Cordell

Academic Editor

PLOS Genetics

Scott Williams

Section Editor

PLOS Genetics

Reviewer's Responses to Questions

**Comments to the Authors:**

Reviewer #1: In this manuscript the authors report their findings following an association study into the genetic factors leading to the individuals to manifest varicose veins with the aim of providing novel target genes, strengthening the evidence of existing associated genes and hopefully provide new directions in treatment development. The authors present an overwhelming set of data in support of theirs findings for an association between genetic variants in genes and VV, although the authors present strong significant association between a number of genes the odds ratios presented are predominantly quite close to 1 and within the confidence intervals (particularly the PIEZO1 findings) suggesting that the likely association is a bit more moderate than what the authors are claiming even within this larger population dataset. I recommend that the authors consider revising their interpretation of strength of the associations detected in their study while addressing the comments below.

Major comments

1) The authors conclude that both PNO1 and PIEZO1 are strongly associated with VV and while their results and interpretation for the PIEZO1 is clearly presented in depth in this manuscript the PNO1 is very briefly mentioned at the start of the results section. I strongly recommend that the authors expand their presentation of the results that support the PNO1 association with VV which can only strengthen their conclusions.

2) On a similar track to the previous comment, the authors should consider presenting their findings for the novel association of TRIM10 and UBE2H in a manner that supports their statements in the conclusion (lines 271 to 276). This section of the conclusion would also benefit from a restructuring of the text to clearly state the novelty of the association of these genes with VV, something that is not immediately clear from the text as is.

3)The statement in the conclusion line 303-303 that the "study demonstrated the association of these genes with endothelial cells and some inflammatory markers..." requires that the authors to clearly elaborate and present this association for the 3 genes not just for the PIEZO1. The authors provide the supporting evidence in additional files, although this is not clear enough to allow a reader of this manuscript to follow the evidence to the author's conclusions.

4) The authors state that one of the strengths of their study is that this "study is the first exome-wide association study of VVs based on a large-scale population". Can the authors claim this as the first such study when other studies like the ones cited by the others in this paper have used the UK Biobank data for association studies with VV and the authors are using a collapsed subset of the data in the UK Biobank. It might be a semantics argument, but it has relevance in a precise scientific unbiased context.

5a) Line 34 Abstract/Line 62 Author Summary/Line102 Introduction - The authors state "...most genes were validated in FinnGen." Can the authors provide more precise information about the level of validation performed in this study here? Also, should it not read "...most variants were validated in FinnGen", my understanding is that FinnGen validates the association of genetic variations with a variety of diseases.

5b) The validation results obtained from FinnGen by the authors should be presented clearly in the main manuscript as evidence to support their claims and not be limited to additional supporting documents and remotely accessible material. If the authors are confident that this validation strengthens their findings it should be presented accordingly. This will also allow the authors to better address the above related comment 5a)

6) As a general comment to the association of LoF variants in the PIEZO1 gene, have the authors considered the predicted loss-of-function tolerance (pLI=0) available from gnomAD and the recessive manner this gene is predicted to act in their interpretation of the association of genetic variation in this PIEZO1 with VV. Together with the predicted highly intolerant to LoF (pLI=0.99) of the ECE1 gene and loss-of-function tolerance (pLI=0) of FBLN7 this may help explain the carriers percentages presented in figure 2b.

7)In Figure 3 the authors present RNAseq data and expression analysis results from datasets obtained from databases detailed in the methods, I would recommend that the authors state clearly the origin of this data in this figure as well or at least refer the potential reader to the methods for the origin of this data.

Minor comments

1) Throughout the manuscript "whole exome sequencing" and "whole-exome sequencing" are used interchangeably, I would recommend the authors pick one form and edit the manuscript accordingly.

2) Line 54 Author Summary/Line 82-83 Introduction - The authors state "Family history is one of the important risk factors for VVs, which suggests that VVs are likely modulated by genetic factors." This is an ambiguous statement to make in my opinion, are the authors referring to a family history of VV or a general family history, also if this is one of the important risk factors what are the other risk factors or is this the most important factor? I recommend that the authors amend this statement to make it clearer.

3) In multiple figure/table legends the abbreviation defined include "Lof, loss of function" when this abbreviation has not been used in the associated figure table. Found in Table 1, Figure 1. Please amend.

4) line 81 - "associated with some health risk factors, including advanced age, female, pregnancy, obesity, high height, and..." consider change to "associated with several risk factors, including advanced age, being female (or sex of the individual), pregnancy, obesity, height, and..."

5) Line 314 - "results did not change obviously after" I recommend the authors revise the usage of "obviously" in this context and rephrasing this sentence in a for a more objective manner.

6) Lines 331-333. The sentence starting "Nevertheless, previous studies using the same..." could be re-ordered for clarity as something like "Nevertheless, previous studies have been able to replicate genetic loci findings for common variants using the phenotypic definitions in GWAS analyses."

7) The text in the Material and Methods section needs to be revised as there are several sentences not conforming to proper grammatical English and too many sentences to individual point out that are poorly constructed in which essential words are missing.

Reviewer #2: The review can be found in the attached file.

Reviewer #3: The authors carried out an association analysis of whole exome

sequencing (WES) data of 350,770 unrelated Caucasian UK BioBank

participants and phenotype data on varicose veins (21,643 cases and

329,127 controls). Previous studies (e.g. Fukaya 2018, Shadrina 2019,

Ahmed 2022 and Levin 2022) used only genotype data which is less

suited to assess rare variants. This current study seems to be the

first large study based on WES, however, genetic studies targeting the

whole exome level are maybe not as unusual as the authors claim.

The authors analysed common and rare variants using single-variant and

gene-based analyses. They identified 34 genes by mapping

single-variant associations to genes, and 3 genes by carrying out

gene-based tests.

The authors identified 19 known and 17 new genes of which most were

validated in FinnGen. Maybe there could be more details given how

the validation was carried out.

The authors also carried out additional analyses of burden

heritability, robustness of the gene-based gene enrichment and Phe-WAS

analyses. The LOVO analysis showed that the association with the

PIEZO1 gene seemed to be influenced by the combination of several rare

variants, while the other two significant genes were rather influenced

by single rare variants.

The main distinction of this study to previous studies is their use of

rare variants. The LOVO analysis is a good start, but could there have

been additional ways to analyse these variants? Rare variants usually

have larger effect sizes, are less likely to be in linkage

disequilibrium with other variants, and could be of greater clinical

relevance.

Maybe variant annotation such as LoF or missense could be added to

Table 1. Is there anything more known about particular effects of the

LoF and missense variants? It would be interesting to understand what

effect the missense variants have on the protein. Additionally, it was

not quite clear which cut-offs were used for the missense annotation.

I am not sure whether Figure 3 supports the claim that the PIEZO1 and

ECE1 genes were expressed in adipose tissue at significantly higher

levels than in other tissues. Figure 3b and 3c seem somewhat

non-specific in terms of tissue enrichment. Additionally, the legend

of Figure 3a does not explain the listed categories.

Gene-SCOUT was used to identify two genes, SCUBE3 and IGF2BP2, that

were negatively correlated to height-related traits similar to

PIEZO1. Was it also used to investigate the other significant genes?

Overall, the text would benefit from some grammar checks and

re-phrasing, and a more structured and more focussed discussion

would be easier to read.

Minor comments:

-------------------

There are some typos in the main text and the legend of Figure 4. Some

references and brief explanations in the main text were missing,

e.g. for FUMA and DAVID. Additionally, a reference is missing for the

claim that a Mendelian randomization study found a positive

correlation between adiposity and the risk of VVs.

Weiner DJ et al. (Nature 2023) appears twice in the references and it

was referenced with a typo in main text 'Winer'.

**Have all data underlying the figures and results presented in the manuscript been provided?**

Reviewer #1: **No: **As mentioned above the authors mention several times that their findings have been validated in/using FinnGen although this supporting evidence is not clearly presented in the manuscript and attached figures/tables. I recommend that this information is made readily available.

Reviewer #2: Yes

Reviewer #3: Yes

PLOS authors have the option to publish the peer review history of their article (what does this mean?). If published, this will include your full peer review and any attached files.

Reviewer #1: No

Reviewe

---

## [Decision Letter · Decision Letter 1]

21 Mar 2024

Dear Dr Yu,

Thank you very much for submitting your Research Article entitled 'Exome sequencing identifies novel genetic variants associated with varicose veins' to PLOS Genetics.

The manuscript was fully evaluated at the editorial level and by independent peer reviewers. The reviewers appreciated the improvements made compared to the previously submitted version, but raised some substantial remaining concerns about the current  version of the manuscript. Based on the reviews, we will not be able to accept this version of the manuscript, but we would be willing to review a much-revised version. We cannot, of course, promise publication at that time.

If you decide to revise the manuscript for further consideration at PLOS Genetics, please aim to resubmit within the next 60 days, unless it will take extra time to address the concerns of the reviewers, in which case we would appreciate an expected resubmission date by email to plosgenetics@plos.org.

We are sorry that we cannot be more positive about your manuscript at this stage. Please do not hesitate to contact us if you have any concerns or questions.

Yours sincerely,

Heather J Cordell

Academic Editor

PLOS Genetics

Scott Williams

Section Editor

PLOS Genetics

Reviewer's Responses to Questions

**Comments to the Authors:**

Reviewer #1: The authors of the manuscript have taken the suggestions and comments from all reviewers to their manuscript. Having for the majority answer them successfully, the authors have now improved their manuscript which following some more minor revisions should be ready for publication.

Minor revision points to consider:

Line 80 - In answering my previous Minor comment 4) the authors have included both my suggestions when my intent was for the authors to pick one (either being female or the sex of individual). I apologize for not being clearer in my original comment but recommend the authors make a minor edit to this sentence one final time.

Line 282 - The portion of the sentence "and not only validated the PIEZO1 it found but additionally identified other genes associated with VV." needs some minor editing to change it into correct grammatical sense.

Line 308 - The authors used the expression "3 rare genes" in this sentence. I question whether the genes are rare or if the variants identified in these genes are rare? Please amend.

Reviewer #2: The review is uploaded as an attachment.

Reviewer #3: The revised manuscript is an improved version compared to the original submission, however there are still many cases of wrong use of tense and ill-phrased terms and sentences.

- Past tense instead of present tense is being used in some sentences added in the revision, e.g.

L114 - The clustering of the VV case and control groups in the 115 principal component analysis was shown in Fig. S1.

L135 - The quantile-quantile (Q-Q) plots of single common 136 variants were shown in Fig. 1b.

L152 - The corresponding Q-Q plots were shown in Additional file 3. Fig.S2.

L383 - The UKB was a large-scale prospective cohort...

Maybe 'offers' instead of 'offered'.

L312 - A study by Backman et al. found that missense variants in PIEZO1 might have a gain-of-function effect and reduce VV risk(34), which offered the possibility of treating VV by pharmacological interventions.

- Probably better described in plural than singular.

L134 - The associations between 32 genes and VV were successfully validated in FinnGen.

L165 - The associations of the three genes we identified in the collapse analysis with VV were further successfully validated in the VV genomic data from FinnGen

L141 - 'Not replicated in FinnGen' rather than 'Not duplicated in FinnGen'

The term 'collapse analysis' sounds slightly odd, especially in the header here.

L 146 - Three VV genes identified by collapse analysis

Maybe better 'rare-variant collapsing analysis' or 'gene- based collapsing approach' or 'variant collapse analysis'.

Whenever the authors write about 'significant level' it should probably be 'significance level', e.g.

L153 - The significant level of the genetic variation associations...

Table 2 - Put definition of 'rare' and 'ultra-rare' variants in Table 2.

Should be clumping instead of clump

L214 - After clump, we identified ...

Does it mean that ECE1 was not associated in the age-matched analysis any longer?

L217 - Collapse analyses were also performed in the matched population and showed that population and showed that PIEZO1 and FBLN7 were still found to be correlated...

Remove 'And' in the following sentence.

L235 - And there were 12 GO terms significantly...

Correct reference should be

L280 - Van Hout et al. study

Maybe better phrase like this.

L308 - The other major finding in this study were the associations of VV with rare variants in three genes (PIEZO1, ECE1, and FBLN7) identified by the variant collapsing analysis.

Probably better singular.

L320 - A mouse model showed that estrogen...

Do the authors mean there were no other cohorts available with VV and WES data?

L363 - Previous WES of VV was lacking,...

Re-phrase the sentence in L367-369.

The cut-offs are missing here.

L421 - Sample-level QC included the removal of samples that were Ti/Tv, were Het/Hom and SNV/indel, had withdrawn their consent, ...

Maybe better 'related' than 'irrelevant'.

L426 - The kinship coefficient threshold of 0.0884 was used to classify secondary kinship as an irrelevant sample.

Maybe better 'higher' than 'more significant'

L428 - kinship coefficients more significant than 0.0884

What procedure was actually used to calculate kinship?

Which cut-offs were used?

L410-413 - A missense variant was considered damaging when it was predicted to be deleterious by SIFT (sorting intolerant from tolerant)(57), PolyPhen-2 HDIV, PolyPhen-2 HVAR(58), LRT (likelihood ratio test)(59) and MutationTaster(60) consistently...

**Have all data underlying the figures and results presented in the manuscript been provided?**

Reviewer #1: None

Reviewer #2: Yes

Reviewer #3: Yes

PLOS authors have the option to publish the peer review history of their article (what does this mean?). If published, this will include your full peer review and any attached files.

Reviewer #1: No

Reviewer #2: No

Reviewer #3: No

---

## [Decision Letter · Decision Letter 2]

7 May 2024

Dear Dr Yu,

Thank you very much for submitting your Research Article entitled 'Exome sequencing identifies novel genetic variants associated with varicose veins' to PLOS Genetics.

The manuscript was fully evaluated at the editorial level and by independent peer reviewers. The reviewers appreciated the attention to an important topic but identified some minor remaining concerns that we ask you address in a revised manuscript.

We therefore ask you to modify the manuscript according to the review recommendations. Your revisions should address the specific points made by each reviewer.

Yours sincerely,

Heather J Cordell

Academic Editor

PLOS Genetics

Scott Williams

Section Editor

PLOS Genetics

Reviewer's Responses to Questions

**Comments to the Authors:**

Reviewer #2: The authors have addressed all the comments and suggested modifications.

I have a few minor suggestions regarding wording before publication:

- In Table 2 legend: 'Number of rare variants: number of markers with MAF < 0.01.'

- l. 189 and 484: 'defined as having a MAF between 1 × 10-5 and 1 × 10-2'

- l. 200: 'of the variant chr1:21227980:A:G'

- l. 214 and methods: can the authors provide the numbers of males and females analyzed?

Reviewer #3: The revised manuscript is much improved, however I still have a few minor comments.

The following expressions should be re-phrased because their use of English or their content do not sound quite correct.

L29. 'a lack of meticulous study on its genetic factors'

L39. 'The pleiotropic of these genes'

L235. 'However, no enrichment results were found due to fewer identified VV genes in males.'

L281. 'be help for therapeutic target discovery'

L291. 'the significance of adipose in the association'

L355. 'association between overlapping gene and VV'

L408. 'whole-exome sequencing on for all participants from the UKB'

I am not clear what is meant by the following statement.

L426. Then we used KING software to calculate pairwise heterozygote concordance rates.

**Have all data underlying the figures and results presented in the manuscript been provided?**

Reviewer #2: Yes

Reviewer #3: Yes

PLOS authors have the option to publish the peer review history of their article (what does this mean?). If published, this will include your full peer review and any attached files.

Reviewer #2: No

Reviewer #3: No

---

## [Decision Letter · Decision Letter 3]

13 Jun 2024

Dear Dr Yu,

We are pleased to inform you that your manuscript entitled "Exome sequencing identifies novel genetic variants associated with varicose veins" has been editorially accepted for publication in PLOS Genetics. Congratulations!

Yours sincerely,

Heather J Cordell

Academic Editor

PLOS Genetics

Scott Williams

Section Editor

PLOS Genetics

Comments from the reviewers (if applicable):

Reviewer's Responses to Questions

**Comments to the Authors:**

Reviewer #2: The authors have addressed all the comments.

Reviewer #3: Thank you for making the final edits to your publication.

**Have all data underlying the figures and results presented in the manuscript been provided?**

Reviewer #2: Yes

Reviewer #3: Yes

PLOS authors have the option to publish the peer review history of their article (what does this mean?). If published, this will include your full peer review and any attached files.

Reviewer #2: No

Reviewer #3: No

**Data Deposition**

http://datadryad.org/submit?journalID=pgenetics&manu=PGENETICS-D-23-01104R3

**Press Queries**

---

## [Editor Report · Acceptance letter]

22 Jun 2024

PGENETICS-D-23-01104R3 

Exome sequencing identifies novel genetic variants associated with varicose veins 

Dear Dr Yu, 

We are pleased to inform you that your manuscript entitled "Exome sequencing identifies novel genetic variants associated with varicose veins" has been formally accepted for publication in PLOS Genetics! Your manuscript is now with our production department and you will be notified of the publication date in due course.

With kind regards,

Zsofia Freund

PLOS Genetics

On behalf of:
